# Higher-order Topological Insulators and Semimetals in Three Dimensions without Crystalline Counterparts

Yu-Feng Mao[1], Yu-Liang Tao[1], Jiong-Hao Wang[1], Qi-Bo Zeng[2], and Yong Xu[1,3*]

[1]*Center for Quantum Information, IIIS, Tsinghua University, Beijing 100084, People's Republic of China*
[2]*Department of Physics, Capital Normal University, Beijing 100048, People's Republic of China and*
[3]*Hefei National Laboratory, Hefei 230088, People's Republic of China*

Quasicrystals allow for symmetries that are impossible in crystalline materials, such as eight-fold rotational symmetry, enabling the existence of novel higher-order topological insulators in two dimensions without crystalline counterparts. However, it remains an open question whether three-dimensional higher-order topological insulators and Weyl-like semimetals without crystalline counterparts can exist. Here, we demonstrate the existence of a second-order topological insulator by constructing and exploring a three-dimensional model Hamiltonian in a stack of Ammann-Beenker tiling quasicrystalline lattices. The topological phase has eight chiral hinge modes that lead to quantized longitudinal conductances of $4e^2/h$. We show that the topological phase is characterized by the winding number of the quadrupole moment. We further establish the existence of a second-order topological insulator with time-reversal symmetry, characterized by a $\mathbb{Z}_2$ topological invariant. Finally, we propose a model that exhibits a higher-order Weyl-like semimetal phase, demonstrating both hinge and surface Fermi arcs. Our findings highlight that quasicrystals in three dimensions can give rise to higher-order topological insulators and semimetal phases that are unattainable in crystals.

Higher-order topological phases represent a significant expansion of conventional first-order topological phases and have experienced considerable advancements in recent years [1–20]. These phases possess edge states of dimension $(n-m)$ $(1 < m \leq n)$ in an $n$-dimensional system, which is in stark contrast to first-order cases that support $(n-1)$-dimensional edge states. For instance, in two dimensions (2D), second-order topological insulators like the quadrupole insulator exhibit four corner states [1, 2]. In three dimensions (3D), second-order topological insulators give rise to four chiral (or helical pairs of) hinge modes [8]. Furthermore, higher-order Weyl semimetals in 3Ds display bulk Weyl nodes that feature both surface and hinge Fermi arcs [21–24].

Apart from crystalline systems, higher-order topological states have also been found in non-crystalline systems such as quasicrystals [25–30], amorphous lattices [31–34] and hyperbolic lattices [35, 36], despite the absence of translational symmetries. Remarkably, these systems in 2Ds can support higher-order topological phases that cannot exist in crystals. For example, a quasicrystal in 2Ds with eight-fold rotational symmetry can possess eight corner modes [25, 26], in stark contrast to crystalline counterparts with two, four or six corner modes [1, 12, 16, 17]. Similar cases occur in amorphous lattices protected by an average symmetry [34]. Although significant progress has been made, it remains unclear whether topological insulators without crystalline counterparts can exist in 3Ds. The challenge lies in establishing the bulk-boundary correspondence in the case of 3D quasicrystals. Previously, a higher-order topological phase in a 2D quasicrystal was characterized by a $\mathbb{Z}_2$ topological invariant determined by the sign of the Pfaffian of a transformed Hamiltonian at high-symmetry

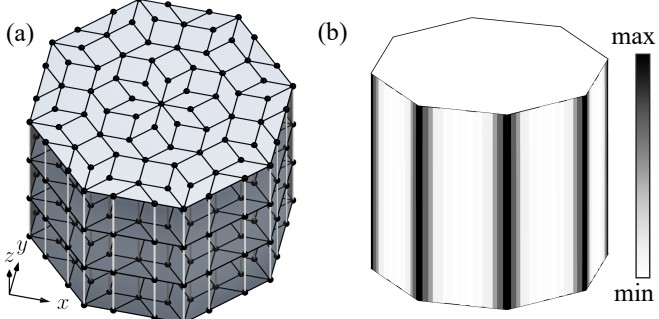

FIG. 1. (a) Schematic illustration of a stack of Ammann-Beenker tiling quasicrystalline lattices on which our tight-binding models are constructed. (b) The zero-energy local density of states (DOS) of the Hamiltonian (1) on a stack of quasicrystalline lattices, illustrating the existence of midgap hinge states.

momenta that is an antisymmetric matrix [25, 26]. However, this invariant cannot be generalized to characterize the chiral hinge modes in 3Ds. Fortunately, the winding number of the quadrupole moment can be employed to characterize the chiral hinge modes when their number is equal to four [19, 32]. Very recently, we have proposed a method to calculate the quadrupole moment in 2D amorphous lattices with eight corner modes [34]. This method thus provides an opportunity to establish the existence of topological phases in 3Ds without crystalline counterparts.

In this work, we theoretically predict the existence of a second-order topological insulator by constructing and exploring 3D model Hamiltonians in a stack of Ammann-Beenker tiling quasicrystals with eight-fold rotational symmetry [see Fig. 1(a)]. We find that there are eight

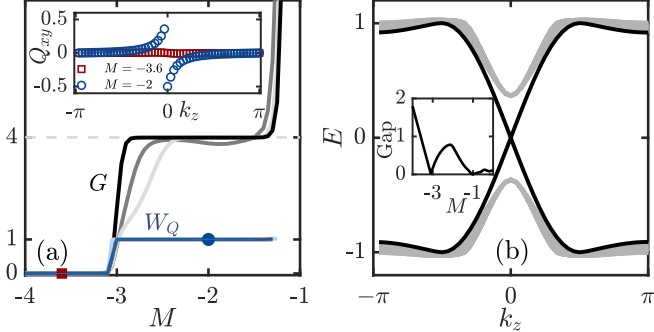

FIG. 2. (a) The longitudinal conductance $G$ (in units of $e^2/h$) along $z$ (gray and black lines) and the winding number of the quadrupole moment $W_Q$ (blue lines) versus $M$ for the Hamiltonian (1). The color lines ranging from light to dark refer to the results for systems with 2377, 4257 and 6449 sites in each plane, respectively. The inset plots the quadrupole moment versus $k_z$ at $M = -3.6$ and $M = -2$. (b) The calculated energy spectrum with respect to $k_z$ for the Hamiltonian (1) with 2377 sites in each plane under open boundary conditions (OBCs) at $M = -2$. The chiral hinge states, which are four-fold degenerate, are highlighted as black lines. The bulk energy gap is shown in the inset. Here, $g = 0.5$.

gapless chiral hinge modes [see Fig. 1(b)] leading to longitudinal conductances of $4e^2/h$. These topological states are not allowed in crystalline materials due to the absence of eight-fold rotational symmetries. To establish the bulk-edge correspondence, we use the method proposed in Ref. [34] to calculate the quadrupole moment and then evaluate its winding number, which confirms the agreement with the observed conductances. Moreover, we show the existence of second-order topological insulators in 3D quasicrystals with time-reversal symmetry (TRS) that support eight helical pairs of hinge states, giving rise to the longitudinal conductance of $8e^2/h$. We find that such a phase is protected by a $\mathbb{Z}_2$ topological invariant defined based on transformed site positions in quasicrystals. Finally, we present a model that showcases the existence of higher-order Weyl-like semimetal phase in 3D quasicrystals. This phase exhibits both hinge Fermi arcs and surface arcs, which are characterized by the quadrupole moment and Bott index, respectively. Notably, unlike higher-order Weyl semimetals in crystals, the quasicrystal exhibits the presence of eight hinge arc states.

*Model Hamiltonian.*— To demonstrate the presence of 3D topological insulators that do not have crystalline counterparts, we stack 2D quasicrystalline lattices to create a 3D lattice as shown in Fig. 1(a) and introduce a tight-binding model on the lattice described by the Hamiltonian

$$\hat{H}_c = \sum_{\boldsymbol{r}}[M\hat{c}^\dagger_{\boldsymbol{r}}\tau_z\sigma_0\hat{c}_{\boldsymbol{r}} + \sum_{\boldsymbol{d}}\hat{c}^\dagger_{\boldsymbol{r}+\boldsymbol{d}}T_c(\hat{\boldsymbol{d}})\hat{c}_{\boldsymbol{r}}], \qquad (1)$$

where $\hat{c}^\dagger_{\boldsymbol{r}} = (\hat{c}^\dagger_{\boldsymbol{r},1}, \hat{c}^\dagger_{\boldsymbol{r},2}, \hat{c}^\dagger_{\boldsymbol{r},3}, \hat{c}^\dagger_{\boldsymbol{r},4})$ with $\hat{c}^\dagger_{\boldsymbol{r},\alpha}$ ($\hat{c}_{\boldsymbol{r},\alpha}$) cre-

ating (annihilating) a particle of the $\alpha$th component at the lattice site of position $\boldsymbol{r}$, and $\tau_\nu$ and $\sigma_\nu$ with $\nu = x, y, z$ are Pauli matrices acting on internal degrees of freedom. The first term describes the on-site mass term, and the second one describes the hopping between two connected sites $\boldsymbol{r}$ and $\boldsymbol{r} + \boldsymbol{d}$ [see Fig. 1(a)] depicted by the hopping matrix $T_c(\hat{\boldsymbol{d}})$ with $\hat{\boldsymbol{d}}$ being the unit vector of $\boldsymbol{d}$. For the intra-layer hopping, $T_c(\hat{\boldsymbol{d}}) = [t_0\tau_z\sigma_0 + it_1\tau_x(\hat{d}_x\sigma_x + \hat{d}_y\sigma_y) + g\cos(p\theta/2)\tau_y\sigma_0]/2$ with $p = 8$ for the Ammann-Beenker tiling quasicrystals and $\theta$ being the polar angle of the vector $\boldsymbol{d}$, and for the interlayer hopping, $T_c(\hat{\boldsymbol{d}}) = (t_0\tau_z\sigma_0 + it_1\tau_x\sigma_z)/2$. Here, $t_0$ and $t_1$ are system parameters, which will henceforth be set to one as the units of energy for simplicity without loss of generality. While the term $g\cos(p\theta/2)\tau_y\sigma_0$ breaks the TRS and eight-fold rotational symmetry, their combination symmetry is preserved, which protects the eight chiral hinge modes. Without the hopping along $z$, the system reduces to a 2D quasicrystal model with eight zero-energy corner modes [25, 26]. The hopping along $z$ clearly breaks chiral symmetry so that the 3D model is not a simple stacking of 2D models.

To map out the phase diagram with respect to the mass $M$, we numerically calculate the zero-temperature two-terminal longitudinal conductance $G$ along the $z$-direction based on the Landauer formula

$$G = \frac{e^2}{h}T(E_F). \qquad (2)$$

Here $T(E_F)$ represents the transmission probability from one lead to the other at the energy $E_F$ for the 3D quasicrystal system connected to two infinite leads along $z$. We calculate the transmission probability $T(E_F)$ at zero energy using the nonequilibrium Green's function method [37–39].

Figure 2(a) displays the numerically computed conductance $G$ as a function of $M$, remarkably illustrating the existence of a region with a quantized value of $4e^2/h$. Specifically, as we increase $M$ from $-4$, the conductance suddenly rises at $M \approx -3.1$, indicating the occurrence of a topological phase transition. In fact, the transition point is associated with the bulk energy gap closing as shown in the inset of Fig. 2(b). In the topological region, we find that the conductance becomes more perfectly quantized at $4e^2/h$ as we enlarge the system size [see Fig. 2(a)], confirming the existence of the topological phase in the thermodynamic limit. To further confirm the origin of the quantized conductance from chiral hinge modes, we plot the energy spectrum of the system at $M = -2$ with respect to the momentum $k_z$ under open boundaries in the $x$ and $y$ directions. The figure clearly shows the presence of gapless chiral hinge modes, which result in the quantized conductance. For each chiral mode, there is four-fold degeneracy due to the $C_8T$ symmetry, the combination of the eight-fold rotational and the time-reversal operations. Such hinge states are

not allowed in a crystal since the eight-fold rotational symmetry is not permitted in a crystalline lattice.

We now propose using the winding number of the quadrupole moment with respect to $k_z$ to establish the bulk-edge correspondence for the higher-order topological insulator,

$$W_Q = \int_0^{2\pi} dk_z \frac{\partial Q_{xy}(k_z)}{\partial k_z}, \quad (3)$$

where $Q_{xy}(k_z)$ is the quadrupole moment at $k_z$. The winding number has been used to characterize the higher-order topological insulator with *four* chiral hinge modes [19, 32]. The quadrupole moment is defined as [40–43]

$$Q_{xy}(k_z) = [\frac{1}{2\pi} \operatorname{Im} \log \det(U_o^{\dagger} \hat{D} U_o) - Q_0] \mod 1, \quad (4)$$

where $U_o = (|\psi_1\rangle, |\psi_2\rangle, \ldots, |\psi_{N_{occ}}\rangle)$ is a matrix consisting of occupied eigenstates of the first-quantization Hamiltonian at $k_z$ under periodic boundary conditions [44], $\hat{D} = \operatorname{diag}\{e^{2\pi i x_l y_l / L^2}\}_{l=1}^{2N_{occ}}$ with $(x_l, y_l)$ being the real-space coordinate of the $l$th degree of freedom, and $Q_0$ is contributed by the background positive charge distribution. If we use the original real-space coordinates of the quasicrystal lattice to calculate the quadrupole moment, we always obtain zero results as clarified in Ref. [34] for the amorphous case. To obtain the reliable quadrupole moment, we perform the transformations of site positions from $(x_l, y_l)$ to $(x'_l, y'_l)$ so that a half or one and a half of a quadrant of sites are transformed into a quadrant [34, 44]. While $\hat{D}$ and $Q_0$ are changed accordingly, we use the same bulk states $U_o$ to evaluate the quadrupole moment as well as its winding number.

Figure 2(a) shows the winding number $W_Q$ with respect to $M$, which exhibits the quantized value of one in the topological regime and zero in the trivial regime, thereby establishing the bulk-edge correspondence of the 3D quasicrystal state. For clarity, we also display the quadruple moment as a function of $k_z$ at $M_z = -2$ and $M_z = -3.6$, illustrating the presence and absence of the winding, respectively.

*Model with TRS.*— We now construct a model with TRS incorporating eight degrees of freedom per site described by the Hamiltonian

$$\hat{H}_h = \sum_{\boldsymbol{r}} M \hat{c}_{\boldsymbol{r}}^{\dagger} \tau_z s_0 \sigma_0 \hat{c}_{\boldsymbol{r}} + \sum_{\boldsymbol{d}} \hat{c}_{\boldsymbol{r}+\boldsymbol{d}}^{\dagger} T_h(\hat{\boldsymbol{d}}) \hat{c}_{\boldsymbol{r}}, \quad (5)$$

where $\hat{c}_{\boldsymbol{r}}^{\dagger} = (\hat{c}_{\boldsymbol{r},1}^{\dagger}, \ldots, \hat{c}_{\boldsymbol{r},8}^{\dagger})$ with $\hat{c}_{\boldsymbol{r},\alpha}^{\dagger}$ creating a fermion of the $\alpha$th component at site $\boldsymbol{r}$, and $\{s_\nu\}$ with $\nu = x, y, z$ is another set of Pauli matrices besides $\{\sigma_\nu\}$ and $\{\tau_\nu\}$. For the intra-layer hopping, $T_h(\hat{\boldsymbol{d}}) = [t_0 \tau_z s_0 \sigma_0 + it_1 \tau_x s_0 (\hat{d}_x \sigma_x + \hat{d}_y \sigma_y) + g \cos(4\theta) \tau_y s_y \sigma_0]/2$ and for the inter-layer hopping along $z$, $T_h(\hat{\boldsymbol{d}}) = (t_0 \tau_z s_0 \sigma_0 +$

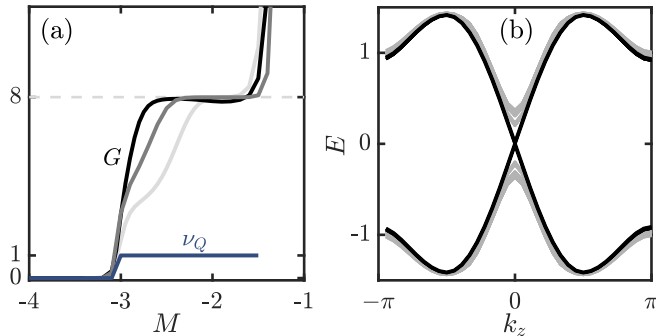

FIG. 3. (a) The longitudinal conductance $G$ (in units of $e^2/h$) (gray and black lines) and the topological invariant $\nu_Q$ (blue line) for the Hamiltonian with TRS in Eq. (5). The gray to black lines correspond to systems with 1137, 2377 and 4257 sites in the $(x, y)$ plane. (b) The energy spectrum versus $k_z$ for the Hamiltonian with TRS with 2377 sites in each plane under OBCs at $M = -2$. The helical hinge states, which are eight-fold degenerate, are highlighted as black lines. Here, $g = 0.5$.

$it_1 \tau_x s_0 \hat{d}_z \sigma_z + it_3 \hat{d}_z \tau_y s_x \sigma_0)/2$. The Hamiltonian now respects the TRS. Similar to the case without TRS, we set $t_0 = t_1 = t_3 = 1$ as the units of energy.

Previously, we develop a $\mathbb{Z}_2$ invariant to characterize the higher-order topology in an amorphous system with TRS supporting four helical pairs of hinge modes in Ref. [32]. The topological invariant $\nu_Q$ is defined based on

$$(-1)^{\nu_Q} = \frac{\operatorname{Pf}[A(\pi)]}{\operatorname{Pf}[A(0)]} \sqrt{\frac{\det[A(0)]}{\det[A(\pi)]}}, \quad (6)$$

where $\operatorname{Pf}[\cdot]$ denotes the Pfaffian of an antisymmetric matrix and $A(k_z) \equiv U_o(-k_z)^{\dagger} \hat{D} T U_o(k_z)$ with $T = i\sigma_y \kappa$, $U_o(k_z) = (|\psi_1(k_z)\rangle, |\psi_2(k_z)\rangle, \ldots, |\psi_{N_{occ}}(k_z)\rangle)$ being a matrix consisting of occupied eigenstates of Hamiltonian (5) at the momentum $k_z$. Similar to the case without TRS, we need to perform the transformation of site positions for the $\hat{D}$ matrix to evaluate the topological invariant.

In Fig. 3(a), we plot the zero-temperature longitudinal conductance $G$ and the $\mathbb{Z}_2$ topological invariant. The figure clearly illustrates the existence of a topological regime identified by the quantized conductance of $8e^2/h$ and the quantized nontrivial value of the topological invariant. The conductance is attributed to the eight helical pairs of hinge modes as shown in Fig. 3(b), which are degenerate due to the $C_8 s_x$ symmetry.

When $t_3 = 0$, the Hamiltonian (5) commutes with $s_y$ so that it can be written as a direct sum of two copies of the Hamiltonian (1) with opposite signs of $g$. We therefore can calculate the winding number of the quadrupole moment in each subspace using the transformed site positions so as to evaluate the spin quadrupole moment

winding number [32] to characterize the system's topology. In fact, despite the absence of the symmetry when $t_3$ is nonzero, we can still compute the spin winding number and find that the results coincide with the $\mathbb{Z}_2$ invariant.

*Higher-order Weyl-like semimetal.*— We now proceed to introduce a model in the stack of 2D quasicrystals that exhibits both the first-order surface modes and second-order hinge modes. The Hamiltonian reads

$$\hat{H}_W(M) = \sum_{\boldsymbol{r}} \hat{c}_{\boldsymbol{r}}^{\dagger} T_0 \hat{c}_{\boldsymbol{r}} + \sum_{\boldsymbol{d}} \hat{c}_{\boldsymbol{r}+\boldsymbol{d}}^{\dagger} T_W(\hat{\boldsymbol{d}})\hat{c}_{\boldsymbol{r}}, \qquad (7)$$

where $T_0 = M\tau_z\sigma_0 + t_c\tau_0\sigma_z$ and $T_W(\hat{\boldsymbol{d}})$ is the hopping matrix that reads $T_W(\hat{\boldsymbol{d}}) = [t_0\tau_z\sigma_0 + it_1(\cos\theta\tau_x\sigma_x + \sin\theta\tau_x\sigma_y) + g\cos(4\theta)\tau_y\sigma_0]/2$ for the intra-layer hopping and $T_W(\hat{\boldsymbol{d}}) = t_0\tau_z\sigma_0/2$ for the inter-layer hopping. Similar to the previous cases, we set $t_0 = t_1 = 1$. The inter-layer hopping changes the mass $M$ to $M + t_0\cos k_z$. Therefore, we can view the system as a stack of 2D systems on quasicrystalline lattices with the mass controlled by $k_z$. Without $t_c$, the chiral symmetry $\Gamma = \tau_x\sigma_z$ (for the first-quantization Hamiltonian) is preserved so that each slice of a system at a fixed $k_z$ cannot develop quantum anomalous Hall insulating phase. In fact, in this case, the system can develop a four-fold degenerate point at the transition point between a normal insulator and a quadrupole insulator as $k_z$ varies, similar to the Dirac point in the regular case [45–47]. To generate the Weyl-like semimetal phase, we add the term $t_c\tau_0\sigma_z$ to break the chiral symmetry, but leave the particle-hole symmetry $\Xi = \tau_x\sigma_x\kappa$ preserved. As a result, both the quadrupole insulator and quantum anomalous Hall insulator can exist, similar to the higher-order Weyl semimetal [21, 22].

Indeed, adding the $t_c$ term splits each four-fold degenerate point into two twofold degenerate ones. The split regions develop the quantum anomalous Hall insulating phases characterized by the Bott index as shown in Fig. 4(a). Here, the Bott index is defined as [48]

$$\text{Bott} = \frac{1}{2\pi} \text{Im Tr} \log(U_y U_x U_y^{\dagger} U_x^{\dagger}), \qquad (8)$$

where $U_x$ and $U_y$ are given by $U_o^{\dagger} e^{2\pi i\hat{x}/L_x} U_o$ and $U_o^{\dagger} e^{2\pi i\hat{y}/L_y} U_o$, respectively, where $\hat{x}$ and $\hat{y}$ are position operators. Apart from the first-order topological phases, the middle region with $|k_z| \lesssim 1.2$ corresponds to the quadrupole insulator with the quadrupole moment of 0.5, which is calculated using the transformed site positions. In this region, there exist eight hinge states at zero energy as shown in Fig. 4(b).

To illustrate the real-space distribution of the hinge and surface states, we plot the $k_z$-resolved local DOS at zero energy in Fig. 4(c). We see that the midgap states in the middle region in $k_z$ are mainly spatially localized on the hinges, while the states in the anomalous Hall region are localized on the surfaces. In contrast to the higher-order Weyl semimetal in a crystal [21, 22], this phase

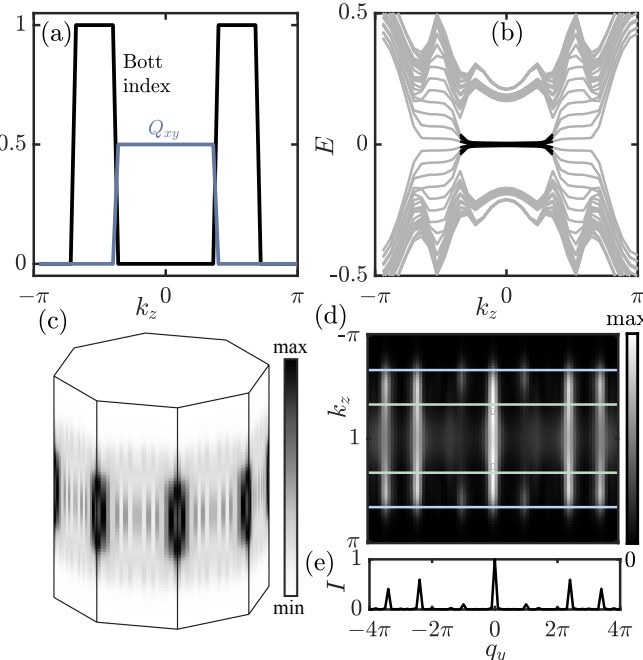

FIG. 4. (a) The quadrupole moment $Q_{xy}$ (blue line) and the Bott index (black line) versus $k_z$ for the Hamiltonian (7). (b) The energy spectrum with respect to $k_z$ for the Hamiltonian (7) under OBCs in the $(x, y)$ plane. The black lines represent the eight-fold degenerate zero-energy hinge states. (c) The $k_z$-resolved local DOS at zero energy, illustrating the existence of hinge states and surface states at different $k_z$. (d) The spectral DOS at the $x$-normal boundaries versus the Fourier momentum $q_y$ calculated by the kernel polynomial method, showing that Fermi arcs connect two degenerate points, whose positions are indicated by the light blue and green lines. (e) The structure factor in Fourier space $q_y$ showing that the positions of Bragg peaks agree precisely with those of Fermi arcs in $q_y$. Here, we consider a system with 2377 sites in the $(x, y)$ plane and $g = 0.5$.

exhibits the local DOS with eight-fold rotational symmetry as enforced by the $C_8T$ symmetry of the Hamiltonian. Our phase is also different from the first-order quasicrystal Weyl-like semimetal with only anomalous surface states [49]. We therefore establish the bulk-edge correspondence for a higher-order Weyl-like semimetal in a stack of 2D quasicrystals without crystalline counterparts.

To further reveal Fermi arcs arsing from surface and hinge states, we calculate the spectral DOS at the energy $E$ [49], $\rho(x, q_y, E, k_z) = \sum_{\boldsymbol{r}\in S_x}\langle q_y, \boldsymbol{r}, \alpha|\delta(E - H_W(k_z))|q_y, \boldsymbol{r}, \alpha\rangle$, where $S_x$ denotes the set of surface sites on a $x$-normal surface, and $|q_y, \boldsymbol{r}, \alpha\rangle$ is the plane wave with the momentum $q_y$. The quantity measures the DOS of the system that an incident plane wave of energy $E$ can couple to on the surface corresponding to an angle-resolved diffraction measurement [49]. Figure 4(d) shows the spectral DOS at zero energy, illustrating the appearance of Fermi arcs of varying intensities with re-

spect to $q_y$ that connect the projections of two degenerate points at $k_z \approx 2.2$. The spectral DOS distribution corresponds to the Bragg peaks of the structure factor [49] $I(q_y) \propto \sum_{q_x} |\sum_{\boldsymbol{R}} e^{i\boldsymbol{q}\cdot\boldsymbol{R}}|^2$ with lattice sites $\boldsymbol{R}$ and the Fourier momentum $\boldsymbol{q} = (q_x, q_y)$. Figure 4(e) shows that the positions of the Bragg peaks in $q_y$ agree perfectly with those of the Fermi arcs in Fig. 4(d). See also the Supplemental Material for the bulk spectral DOS revealing the existence of the Weyl-like degenerate points [44].

In summary, we have demonstrated the existence of 3D second-order topological insulators and higher-order Weyl-like semimetals in a stack of 2D quasicrystals with eight-fold rotational symmetry. We have also established the bulk-edge correspondence in these systems. Importantly, it is worth noting that such phases cannot appear in a 3D crystal due to the absence of eight-fold rotational symmetry. Moreover, our findings can be generalized to other quasicrystals, such as Stampfli-tiling quasicrystals with twelve-fold rotational symmetry. While our analysis focuses on the case with translational symmetry along the $z$-direction, the implications of our work may be significant for fully 3D quasicrystalline systems. Additionally, these intriguing phases could potentially be experimentally observed not only in realistic quasicrystalline materials but also in metamaterials, such as phononic [50], photonic [51], electric circuit [52] and microwave systems [53].

*Note added*: Recently, we became aware of a related work [54] where higher-order Dirac semimetals in 3D quasicrystals have been studied.

The work is supported by the National Natural Science Foundation of China (Grant No. 11974201) and Tsinghua University Dushi Program.

———————

* yongxuphy@tsinghua.edu.cn

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

In the Supplemental Material, we will show in detail how the periodic boundary conditions (PBCs) and site position transformations are realized in Section S-1 and provide the bulk spectral DOS in Section S-2.

## S-1. DETAILS ON CONSTRUCTIONS OF PERIODIC BOUNDARY CONDITIONS AND SITE POSITION TRANSFORMATIONS

In the main text, we have applied PBCs in the $(x, y)$ plane to calculate the quadrupole moment and the Bott index. Figure S1 shows how the PBCs are realized in the Ammann-Beenker tiling quasicrystalline lattices. Specifically, the sites at a boundary are connected to the sites at the other boundary. For example, the sites $B$ and $C$ at a boundary are connected to the sites $B'$ and $C'$ at the other boundary, respectively. The corner sites are connected to the other two corner sites, e.g., the corner site $A$ is connected to both the sites $A'$ and $A''$.

In the main text, we use the winding number of the quadrupole moment and the $\mathbb{Z}_2$ topological invariant to characterize the topological phases. Their calculations involve the evaluation of the $\hat{D}$ matrix dependent on the site positions. If we use original lattice site positions in a quasicrystal, we obtain zero quadrupole moment [see Fig. S1(b)]. To obtain the reliable quadrupole moment, we perform the transformation of site positions [1]. Specifically, we move the sites in the light green (blue) region in Fig. S1(c) to the first or third (second or fourth) quadrant in Fig. S1(d) while keeping its distance from the center unchanged. After that, we modify the $\hat{D}$ matrix accordingly.

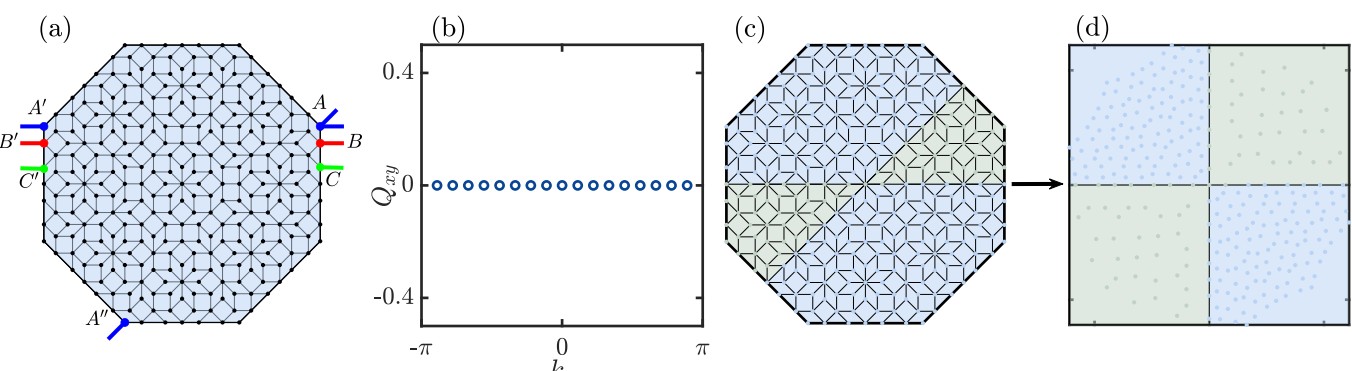

FIG. S1. (a) Schematic illustration of how the PBCs are realized in the Ammann-Beenker tiling quasicrystalline lattices. Examples show that the sites $B$ and $C$ at a boundary are connected to the sites $B'$ and $C'$ at the other boundary, respectively. The corner site $A$ is connected to both the sites $A'$ and $A''$. (b) The quadrupole moment $Q_{xy}$ as a function of $k_z$ for the Hamiltonian (1) in the main text at $M = -2$ calculated using the original lattice site positions in a quasicrystal. We see that the quadrupole moment always vanishes. (c)-(d) Schematic illustration of how the lattice site positions are transformed from (c) to (d) to obtain a reliable quadrupole moment. Specifically, for a lattice site with the polar angle of $\theta$ inside the quasicrystalline lattice, if $m\pi \leq \theta \leq m\pi + \pi/4$ ($m = 0$ or 1), then we change the angle to $\theta_d = m\pi + 2(\theta - m\pi)$, and if $m\pi + \pi/4 \leq \theta \leq (m+1)\pi$, then we change it to $\theta_d = [2\theta + (m+1)\pi]/3$.

## S-2. WEYL-LIKE POINTS REVEALED BY THE BULK SPECTRAL DOS

In the main text, we have shown that the Fermi arcs appear in the spectral DOS resolved in the Fourier momentum $q_y$ when we consider surface sites as the coupling sites. In this section, we provide the spectral DOS at zero energy by considering the bulk sites as the coupling sites in Fig. S2 and find that the Weyl-like degenerate points manifest in the spectral DOS as bright spots. The figure also illustrates that the positions of Bragg peaks of the structure factor in $q_y$ agree perfectly with those of Weyl-like points.

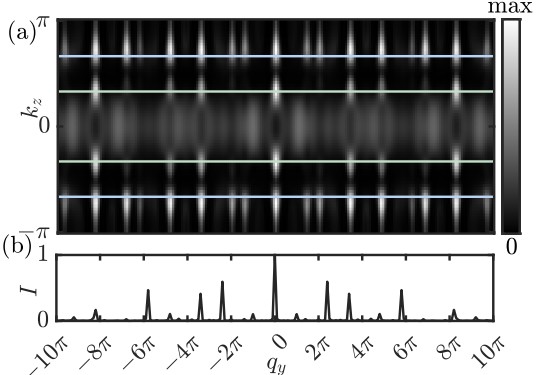

FIG. S2. (a) The bulk spectral DOS at zero energy calculated by considering the bulk sites as the coupling sites. It is calculated in a quasicrystalline lattice with 2377 sites. (b) The structure factor in Fourier space $q_y$.

* yongxuphy@tsinghua.edu.cn

[1] Y.-L. Tao, J.-H. Wang, and Y. Xu, arXiv: 2306.02246 (2023).