# Peer review of "Higher-order Topological Insulators and Semimetals in Three Dimensions without Crystalline Counterparts"

_SciPost Physics_

## Round 2 · Referee Report · Daniel Varjas (Referee 1) · 2024-3-7

Strengths

The manuscript is clearly and succintly written, addresses an open problem about the existence of quasicrystalline higher-order TI's in 3D and higher-order Weyl semimetals.

Weaknesses

  • The manuscript has a large overlap with other works, especially (as the authors also note) with "Quasicrystalline second-order topological semimetals", Phys. Rev. B 108, 195306 (2023)).

  • Some of the information presented in the figures is unclear, see Requested changes for details.

Report

The manuscript presents studies of 3 dimensional quasicrystalline higher-order topological insulating and semimetallic phases. The presentation is clear, and the autors present a thorough analysis of simple tight-binding models.

While the manuscript is of high quality, I have to conclude that the novelty criteria of Scipost Physics are not met. This is mainly because of the large overlap with Phys. Rev. B 108, 195306 (2023), whose first arxiv version is dated a few weeks earlier than this manuscript's.

In conclusion, I recommend publication of the manuscript in SciPost Physics Core after the minor clarification requests below are addressed.

Requested changes

  • In Fig. 1. b) the geometry is somewhat unclear. Wether this is a section of an infinite slab in the z direction, or a mixed real-and-momentum-space (x, y, k_z) figure like Fig. 4. c), this should be clarified. This also raises the question, what happens in a finite slab geometry? As chiral hinge modes cannot simply terminate, they have to continue in some fashion on the top and bottom of the slab. It could be instructive to also show this.

  • In Fig. 4. c) a k_z axis label should be added to clarify the nature of the plot. In the current plot it is unclear whether the surface states coexist with the hinge states for the intermediate k_z regime around 0. From the Bott index analysis, this range should only have hinge states as far as I understand. The authors should enhance the quality of the plot and add further explanation to clarify this question.

  • In the text the abbreviation "3Ds" is used, I believe "3D" is more common even when abbreviating "3 dimensions".

---

## Round 2 · Referee Report · Anonymous (Referee 2) · 2024-4-4

Strengths

The manuscript studies different topological phases in 3D systems that are obtained by stacking 2D Ammann-Beenker tilings.

Weaknesses

  1. The manuscript is quite densely written, requiring a lot of prior knowledge and thus somewhat unapproachable to beginners in the field. Since SciPost does not impose a text limit, I believe that adding further details to the manuscript would enrich it and make it more self-contained.

  2. Higher-order topology in 2D quasicrystalline settings was studied in great detail, see e.g., PhysRevLett.123.196401 (2019), PhysRevLett.124.036803 (2020), PhysRevB.104.245302 (2021), Communications Physics 4, 108 (2021). It is therefore not surprising for me that appropriately stacking such systems could realize a topological phase with chiral/helical hinge modes that are protected by the symmetries of an underlying quasicrystal. In fact, since hinge modes of higher-order topological insulators are known to survive in amorphous settings, see Phys. Rev. Lett. 126, 206404 (2021), it is quite likely for them to exist in 3D quasicrystals studied in this work. Furthermore, several recent works PhysRevB.108.L121109 (2023) and PhysRevB.108.L121109 (2023) [Ref. 54 of the manuscript] have studied first- and second-order Weyl semimetal phases in 3D systems obtained by stacking 2D quasicrystals. For these reasons, I believe the manuscript lacks sufficient novelty for a publication in SciPost Physics and find it more appropriate for SciPost Physics Core.

Report

The authors thoroughly study higher-order topological insulating and semimetallic phases in 3D quasicrystals. They distinguish these phases by appropriate topological invariants, and calculate the observable like e.g., the electrical conductance for HOTI phases. The results are sound and quite timely given the recent interest in topology of non-crystalline systems. They represent a nice addition to the field of topological phases in quasicrystalline systems; however, I do not find them novel enough to grant a publication in SciPost Physics. I believe the manuscript can be published in SciPost Physics Core after minor comments are addressed (see the report).

Requested changes

  1. The authors should give expressions for all relevant (local and nonlocal) symmetries for each of the three different topological phases. For a HOTI phase with broken time-reversal, I find the discussion in the top right paragraph a bit confusing. In particular, the authors state "While the term g cos (pθ/2) τyσ0 breaks the TRS and eight-fold rotational symmetry, their combination symmetry is preserved, which protects the eight chiral hinge modes. Without the hopping along z, the system reduces to a 2D quasicrystal model with eight zero-energy corner modes [25, 26]. The hopping along z clearly breaks chiral symmetry so that the 3D model is not a simple stacking of 2D models." According to my understanding, setting $g \neq 0$ and the hopping along the z direction $it_1 \tau_x \sigma_z = 0$ or the other way around ($g=0$ and nonvanishing hopping along the z direction) preserves all three local symmetries. Only when both terms are simultaneously nonzero are the time-reversal and chiral symmetries broken, while the particle-hole symmetry is preserved. I find this not to be clear in the manuscript. For a HOTI phase protected by the TRS, what is a $C_8 s_x$ symmetry mentioned on page 3, and its symmetry constraint?

  2. Concerning the calculations of electrical conductance, it is not clear to me whether for its calculation was used a system that is finite or infinite along the z direction. To my understanding, in the caption of Fig 2 are mentioned only system sizes in xy plane. Perhaps this can be clarified by illustrating the transport setup in e.g., Figure 1. Furthermore, I cannot find what are the system sizes used to produce Figure 1b and the inset of Figure 2a. Finally, this is a suggestion, but I think that Fig. 2a (Fig. 3a) could be improved by defining two vertical axes on left and right sides for the conductance and the winding number (topological invariant $\nu_Q$), respectively. In the current plots, the changes of the topological invariants with system size are barely seen.

  3. Can the authors explain their statement on page 1: "However, this invariant cannot be generalized to characterize the chiral hinge modes in 3Ds."?

  4. On page 3 in the discussion bellow Eq. (4) is written: “To obtain the reliable quadrupole moment, we perform the transformations of site positions from $(x_l, y_l)$ to $(x′_l, y′_l)$ so that a half or one and a half of a quadrant of sites are transformed into a quadrant [34, 44].” This is a crucial step for calculating the invariant for all three topological phases, yet I find it very unlikely that a reader will gain any understanding about what is done from this sentence. Perhaps an illustration would be beneficial. Furthermore, since the authors comment on having a vanishing quadrupole moment when using original coordinates, could they explain in couple of sentences why doing the position transformation makes it possible to retrieve nontrivial topology? To my understanding, this is because the transformation makes an odd number of corner modes occupy one quadrant.

---

## Editorial Decision

awaiting_resubmission